# Real-Time Tweet Analytics Using Hybrid Hashtags on Twitter Big Data Streams †

**Vibhuti Gupta** * 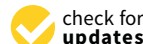 **and Rattikorn Hewett** *

Department of Computer Science, Texas Tech University, Lubbock, TX 79415, USA
* Correspondence: vibhuti.biet06@gmail.com (V.G.); rattikorn.hewett@ttu.edu (R.H.)
† This paper is an extended version of our presentation in 2018 IEEE International Conference on Big Data, Seattle, WA, USA, 10–13 December 2018.

**Abstract:** Twitter is a microblogging platform that generates large volumes of data with high velocity. This daily generation of unbounded and continuous data leads to Big Data streams that often require real-time distributed and fully automated processing. Hashtags, hyperlinked words in tweets, are widely used for tweet topic classification, retrieval, and clustering. Hashtags are used widely for analyzing tweet sentiments where emotions can be classified without contexts. However, regardless of the wide usage of hashtags, general tweet topic classification using hashtags is challenging due to its evolving nature, lack of context, slang, abbreviations, and non-standardized expression by users. Most existing approaches, which utilize hashtags for tweet topic classification, focus on extracting hashtag concepts from external lexicon resources to derive semantics. However, due to the rapid evolution and non-standardized expression of hashtags, the majority of these lexicon resources either suffer from the lack of hashtag words in their knowledge bases or use multiple resources at once to derive semantics, which make them unscalable. Along with scalable and automated techniques for tweet topic classification using hashtags, there is also a requirement for real-time analytics approaches to handle huge and dynamic flows of textual streams generated by Twitter. To address these problems, this paper first presents a novel semi-automated technique that derives semantically relevant hashtags using a domain-specific knowledge base of topic concepts and combines them with the existing tweet-based-hashtags to produce Hybrid Hashtags. Further, to deal with the speed and volume of Big Data streams of tweets, we present an online approach that updates the preprocessing and learning model incrementally in a real-time streaming environment using the distributed framework, Apache Storm. Finally, to fully exploit the batch and stream environment performance advantages, we propose a comprehensive framework (Hybrid Hashtag-based Tweet topic classification (HHTC) framework) that combines batch and online mechanisms in the most effective way. Extensive experimental evaluations on a large volume of Twitter data show that the batch and online mechanisms, along with their combination in the proposed framework, are scalable, efficient, and provide effective tweet topic classification using hashtags.

**Keywords:** Twitter; Hybrid Hashtags; Big Data stream; ontology; Apache Storm

## 1. Introduction

In today's digital era, social media platforms generate huge volumes of data with high velocity and variety (i.e., images, text, and video). Every day, approx. 900 million photos are uploaded on Facebook, 500 million tweets are posted on Twitter, and 0.4 million hours of videos are uploaded on YouTube [1], causing potentially unbounded and ever-growing Big Data streams. Twitter, a popular microblogging social media platform, is widely used today by millions of people globally to remain socially connected or obtain information about worldwide events [2,3], natural disasters [4], healthcare [5], etc. Twitter

users act as distributed "social sensors" which report happenings globally [4]. The short text messages used in Twitter for communication are known as tweets, which can be up to 280 characters of length. A user can follow other users, read their tweets, and also repost the tweets, which are known as retweets. Due to this large deluge of data being generated by millions of Twitter users every day, tweet analytics is viewed as a fundamental problem of Big Data streams. Automated detection of topics being discussed in tweets using tweet analytics techniques is extremely useful in certain cases; for instance, detecting real-time events such as an earthquake [4], predicting needs of people during natural disasters such as Hurricane Harvey in Houston [6], and detecting influenza epidemics [5].

Hashtags, user-defined hyperlinked words, act as a convention for adding additional context and metadata to tweets and are extensively used in Twitter [7]. They are used to indicate the topic of the tweet and facilitate the searching of tweets with a common subject. Hashtags cover an immense range of topics and user interests such as #health, #graduation, #presedentialelection, #angry, #Oscars2016, and #HarveyRescue, used to convey a health state, topic, political campaigns, emotion, event, disaster rescue, etc.

Recent research in tweet analytics has studied how hashtags can be effectively applied for determining peak time popularity [8,9], tweet retrieval [10–12], and tweet keyword extraction [13]. While many hashtag applications are successful, tweet topic classification using hashtags remains challenging due to the evolving nature of tweets and hashtags. Moreover, tweets are very noisy with the combination of slang, abbreviations, emoticons, URLs, and ambiguous words, which make the extraction of relevant information an arduous task. To make things worse, there is no standard on how hashtags are created or expressed. The same subject can be expressed with different hashtags at different points of time (e.g., #omg, #ohmygod) by a user. Typically, users use multiple topic hashtags in a tweet that makes the task of identifying relevant hashtags for a specific topic harder. For example, if a user tweets "Enjoyed Friday with #movie, #hollywood, #food, #restaurant, #art, #sketch", this tweet shows the user's intent that he enjoyed Friday by watching a movie, going to restaurant, and drawing a sketch. This tweet discusses multiple topics, i.e., entertainment, food, and art, which makes it hard to determine a specific topic of discussion.

The majority of existing approaches [14–23] utilizing hashtags for tweet topic classification are lexicon-based approaches that focus on extracting hashtag concepts using external lexicon resources to derive semantics. However, due to the rapid evolution and casual expression of hashtags, these approaches lack generality and scalability. Lexicon-based approaches are widely used for sentiment analysis [7,14,15,24–28] but not for general topic classes. Some lexicon-based approaches [20–23,29,30] that use existing knowledge bases [31–33] for general tweet topic classification suffer from a lack of hashtag words in their knowledge bases due to the highly variant nature of hashtags, and have complex structures that makes their automation harder and unsuitable for real-time Twitter stream analytics.

Some tweet topic classification studies [34,35] have focused on classifying tweets using a combination of other features with hashtags instead of primarily improving the semantics of hashtags. Research in [34] used social network information, while [35] used a set of features from a user's profile with hashtags to classify tweets into topics.

Another major limitation of existing tweet topic classification approaches [34–36] is that they work in a centralized environment, while real tweet big data stream requires an online and distributed approach to deal with fast and dynamic arrival rates of tweets. Therefore, there is a need to develop tweet topic classification approaches that scale well and provide real-time processing. The research literature includes large-scale implementations for Twitter sentiment analysis [26,27] but not for tweet topic classification. Moreover, the topic classifier must have the capability to update its preprocessing and learning model incrementally so that the classifier automatically adapts to new features and improves accuracy with time.

Due to the large variation in the hashtags, tweet topic classification using batch learning alone is not sufficient since the fast evolution of topics and user interests rapidly make the classifier obsolete, which leads to incorrect predictions. Stream learning can resolve this issue by continuous model

updates over time as new data arrives but using it alone will not be beneficial, due to less supervision in feature selection and high dependence on arriving data for model predictions. Both the batch and online mechanisms can have performance advantages in different situations. For example, batch learning can be beneficial in initially identifying domain-specific hashtags that are representative of a given domain class in tweet topic classification, while stream learning can help in updating the relevant hashtags in real-time as new data arrives. Hence, batch learning can work well with a fixed set of features in a non-evolving environment, while stream learning is best suited to a varying number of features in an evolving environment. This shows that both batch and online mechanisms are applicable in specific problem domains when used alone and lack generality. In addition, the batch classifier will be obsolete due to data evolution over time, while the stream classifier might not improve predictions with initial low-quality features. This issue can be resolved by combining batch and online mechanisms, since the batch component might help in building an initial model with good set of features, while the online component can help improve the model with new sets of features in real-time. This leads to our motivation to propose a framework that exploits both batch and online mechanisms for tweet topic classification using hashtags.

This research aims to exploit the domain-specific knowledge first, to identify a set of strong hashtag predictors (i.e., Hybrid Hashtags) for tweet topic classification [37]. The proposed approach consists of two types of hashtags: 1) those extracted from input tweet data and 2) those derived from a knowledge base of topic (or class) concepts (or topic ontology) by using "Hashtagify" [38], a tool to generate "similar" hashtags from a given term (see more details in [38]). The Hybrid Hashtags approach helps improve accuracy of classifying topic classes using hashtags alone and resolves ambiguity. The effectiveness of this approach is evaluated in a batch environment, i.e., the Hybrid Hashtag-based Tweet topic classification (HHTC)-Batch version of the framework, which shows that our proposed approach is effective in classifying tweet topics as compared to other state-of-the-art approaches. Furthermore, it is general in the sense that it can be applied to any given class concept of any domain.

To deal with the speed and volume of Big Data streams, we developed the Hybrid Hashtags approach for online computation [39]. This provides efficient updates for preprocessing and learning of the model incrementally in real-time processing. This online method helps in automatically updating the Hybrid Hashtags as the new tweet data arrives, and enhances distributed and scalable processing. The effectiveness of this approach is evaluated in a stream environment, i.e., the HHTC-Stream version of the framework. Being deployed in a real-time stream processing framework, Apache Storm [40], we not only evaluate the real-time model updates of our proposed online method, but also its scalability, by throughput and speed experiments.

Because the batch and online mechanisms can have performance advantages in different situations, this paper proposes a comprehensive framework (Hybrid Hashtag-based Tweet topic classification (HHTC) framework) for tweet topic classification using Hybrid Hashtags, which combines batch and online mechanisms in the most effective way in a real-time distributed framework (Apache Storm) [40]. Our proposed framework, HHTC, not only helps improve tweet topic classification by building an initial model with a good set of hashtag features selected through the Hybrid Hashtags approach but also helps in providing a real-time analytics solution by continuously updating those Hybrid Hashtags. Overall, our proposed framework provides a comprehensive solution for tweet analytics using hashtags that first provides a simple and effective technique to extract relevant hashtags for any given class/topic and then employs that technique in an online and distributed manner to deal with large scale Twitter Big Data streams. Extensive experimental evaluations not only show the individual performance advantages of batch and online mechanisms, but also the combined performance in the proposed framework.

The remainder of the paper is organized as follows: Section 2 discusses related work. In Section 3, we describe some of the preliminaries for the approach and the Apache Storm framework. Section 4 presents the proposed tweet topic classification framework with all three versions i.e., batch, streaming,

and a combination of both. Section 5 provides information about the dataset used and its characteristics. Section 6 presents an experimental evaluation and the results. Finally, Section 7 concludes the paper with a discussion and future work.

## 2. Related Work

The last decade has seen an increase of interest in studies of tweet analytics using hashtags. Most existing approaches can be broadly categorized into lexicon-based approaches, sentiment and emotion analytics approaches, tweet retrieval approaches, and hashtag recommendation approaches.

There is a large body of work using lexicon-based approach for sentiment analysis. Lexicon-based approaches are widely studied for conventional texts such as blogs, forums, and reviews [18,41]. However, very few studies focus on analyzing hashtags using the lexicon-based approach. Among those that do, a selection focus on identifying sentiment-bearing or non-sentiment-bearing hashtags using lexicon resources (i.e., dictionary of opinion terms) and use the identified hashtags as features for tweet sentiment classification. Simeon et al. [14,15] applied word segmentation algorithms and combined multiple lexical resources (e.g., AFINN [42], SentiStrength [19], and Bing Liu Lexicon [43]) to identify sentiment- and non-sentiment-bearing hashtags. In their approach, hashtags are first divided into smaller semantic units using a word segmentation algorithm and then combined with different lexical resources to build classification models. Finally, each model is being tested individually and in combination on real Twitter data to evaluate its effectiveness. The majority of works focus on leveraging lexicon resources for the whole tweet textual content [16–19]. Ortega et al. [16] used WordNet [44] and SentiWordNet [45] to identify polarity and performed rule-based classification on Twitter data. Saif et al. [17] proposed a lexicon-based approach (i.e., SentiCircles) that uses co-occurrence patterns of words in different contexts to identify sentiment orientation of words. In the current study, we also analyze hashtags using additional resources (i.e., manually constructed knowledge base) but the difference between the above work and ours is that we focus in identifying general topic-based hashtags and use them for tweet topic classification.

Other research has used external lexical sources such as Wikipedia [31], DBPedia [32], and Freebase [33] to enhance the semantic context of tweets for tweet topic classification [20–23,29,30]. Research in [20,22,23] used Wikipedia to enhance contextual semantics of tweets. Genc et al. [20] used the semantic distance between the tweet textual content and the Wikipedia pages to determine similar pages and categories. Authors in [22,23] used concepts derived from Wikipedia for identifying semantic relatedness of text. Cano et al. [21] utilized multiple knowledge sources (i.e., DBPedia, Freebase, etc.) for detecting the topics in tweets. They used the entities present in the tweets and enriched their contextual semantics by utilizing multiple knowledge sources. Semantic features were derived from this information to improve the Twitter topic classification. These studies utilized external knowledge bases similarly to us but they used the approach to enhance the concepts present in the tweet, while we utilize the concepts derived from a knowledge base to improve the quality of hashtags to be used as features for tweet topic classification. Furthermore, our approach is top-down as opposed to the existing bottom-up approaches.

Hashtags work well for sentiment and emotion classification [24,25,28]. Research in [25] uses hashtags and smileys as sentiment labels to classify diverse sentiment types. Hashtags and smileys are first grouped into five categories, i.e., strong sentiment, most likely sentiment, context-dependent sentiment, focused, and no sentiment, and then various features are used to classify tweets. Mohammad et al. [24] uses emotion word hashtags as labels to create a hashtag emotion corpus for six basic emotions and used it to extract a word–emotion association lexicon. In [28], authors proposed a bootstrap approach to automatically identify the emotion hashtags; they started with a small number of seed hashtags and used them to collect and label tweets with five prominent emotion classes. Then, an emotion classifier was built for every emotion class and applied to a large pool of unlabeled tweets. The hashtags extracted from these unlabeled tweets were scored and ranked to extract high ranked hashtags, which was then used to form an emotion hashtag repository. A key limitation of these

approaches is that they can be easily applied for identifying sentiments since their semantics can be easily captured by a single hashtag (e.g., #sad, #happy) but they are not suitable for identifying general topics (e.g., #health, #entertainment) whose semantics require diverse set of hashtags.

Work on hashtag clustering includes metadata-based [46–48] and text-based contextual semantic clustering [49–51] approaches. Authors in [47] used WordNet and Wikipedia as dictionary metadata sources to identify semantics of hashtags. They used WordNet concepts if the hashtag matched an entry in it, otherwise Wikipedia to identify concept candidates. Then, a similarity matrix was formed between all pairs of hashtags to cluster them. A major drawback of their approach is that they focused on word-level rather than concept-level semantics because clustering results were not effective. To resolve this issue, [46] and [48] used sense-level semantics for clustering hashtags. Research in [49–51] used hashtags present in tweet texts to derive contextual semantics. Work in [52] proposed a hybrid approach combining metadata- and text-based semantic clustering to improve the semantics of hashtags. Song et al. [36] proposed a clustering approach to classify hashtags in various news categories, e.g., entertainment and sports, and then provided a ranking method to extract the top hashtags in each category. Belainine et al. [53] developed an approach to improve the semantics of hashtags by using a combination of knowledge from WordNet to disambiguate the meaning in tweets and reduce synonymous terms. In the current study, we utilize the metadata provided by a knowledge base to derive concepts for the given topic to identify hashtags using the tool Hashtagify, combined with the tweet hashtags to derive contextual semantics. Hence our work focuses on deriving relevant hashtags for the topic using the semantic concepts derived from ontology instead of focusing on extracting hashtag concepts to derive semantics.

Some research [54,55] uses ontology to classify long documents and for Twitter sentiment analysis. In [54], ontology is used to identify appropriate classes for multilabel classification of economics documents in various categories. Work in [55] uses ontology for more elaborate sentiment analysis of Twitter posts, instead of determining sentiment polarity of tweets. None of these approaches [54,54] uses ontology to identify appropriate hashtags for tweet topic classification.

Due to the massive volume of data generated by Twitter, a significant amount of research has been undertaken in designing large-scale systems for sentiment analysis [26,27,56,57]. Research in [26,57] used all hashtags and emoticons as sentiment labels and exploited various pattern, words, and punctuation features to classify tweet sentiment in a distributed manner using MapReduce [58] and Apache Spark [59] frameworks. Work in [27] proposed a large-scale distributed system for real-time Twitter sentiment analysis using MapReduce [58]. They used a lexicon-based approach in which they assigned sentiment polarity according to the sum of sentiment scores of each word in a tweet. The main distinction of the current study work is that we propose completely online and distributed preprocessing and learning approaches. In terms of distributed processing, our processing is completely online, implemented on the Storm framework, as opposed to using MapReduce [58], which is a form of distributed batch processing. Although both Apache Storm and Apache Spark are forms of data stream processing, Apache Storm is an online framework while Apache Spark [59] processes data in micro-batches and is therefore not applicable to our work.

Research in tweet retrieval and hashtag recommendation [49,50,52,60–66] is focused on retrieving relevant tweets and recommending hashtags for the tweets with missing hashtags. Research in [49,52,60,61] primarily focuses on retrieving semantically connected hashtags. Bellaachia et al. [60] exploit hashtags to enhance graph-based key phrase extraction from tweets. They extract topical tweets using latent Dirichlet allocation (LDA) [67] and then auxiliary hashtag tweets from the topical tweets by following the hashtag links. Furthermore, the hashtags from topical and auxiliary hashtag tweets are combined in a lexical graph and a ranking algorithm is applied to rank the keywords for a specific topic. Muntean et al. [49] applied k-means to cluster a large set of hashtags in a distributed environment using MapReduce [58]. Recent work on hashtag recommendation [65] uses word embeddings to recommend hashtags for health-related keywords. In [66], the authors first find similar tweets for a given tweet query and rank hashtags in those tweets for recommendations.

Some recent studies identify topic-relevant hashtags [10] and predict peak time popularity [8] for tweet hashtags. Research in [10] used latent Dirichlet allocation (LDA) [67] first to determine the topic distributions and then a support vector machine (SVM) to divide the tweets according to their relevance to the topic. Finally, all of the topic distributions are associated with their class labels to identify relevant and irrelevant hashtags. This approach utilizes a combination of topic modelling with a supervised machine learning technique to retrieve the relevant hashtags, while the current study utilizes a domain-specific knowledge base for relevant hashtags retrieval.

## 3. Background

### 3.1. Hybrid Hashtags Approach

The concepts for a specific domain topic that help to describe the semantics of the topic are termed domain-specific concepts. Identifying the domain-specific concepts and representing them in a graphical representation (where each node represents a concept and each link represents a relationship between them) is known as a domain-specific knowledge base or ontology. For example, a sports topic can involve many concepts, such as sports type, teams, and sports events. Ontologies have been widely used in the past for natural language generation, extracting semantics from unstructured texts, and intelligent information integration [55]. Here, we apply it to tweet hashtags to extract domain-specific or ontology-driven hashtags to cover various aspects of a general topic/class.

Our Hybrid Hashtags approach [37] starts by building a manual ontology/knowledge base of each given class/topic (e.g., entertainment, sport). Each of the terminal node concepts is fed into an automated tool, Hashtagify [38], to retrieve a set of concept-based hashtags. These concept-based hashtags are ordered in decreasing order of correlation with the given concept. The correlation defines the semantic relatedness of hashtags and controlled by the specified parameter $k$. The lower the values of $k$, the more the retrieved set of hashtags is semantically relevant. The correlation score [39] between the hashtag $h$ and concept $c$ can be computed using Equation (1), where $c$ and $h$ are vectors representing the frequency of co-occurrence of concept $c$ and hashtag $h$ in the Hashtagify data while $\bar{c}$, $\bar{h}$, $S_c$, $S_h$ are the means and standard deviations of the values, respectively.

$$corr(c, h) = \frac{\sum_{i=1}^{n}(c_i - \bar{c})(h_i - \bar{h})}{(n-1)S_c S_h} \tag{1}$$

For example, if the value of $k$ is 4 then 4-correlated hashtags retrieved for hashtag #xbox are #PS4, #Amazon, and #XboxOne. This process is repeated with all terminal node concepts to retrieve a set of $k$-correlated hashtags known as ontology-driven hashtags. Then, $k$-correlated hashtags for these ontology-driven hashtags are retrieved from the data, which are known as tweet-based-hashtags. The combination of both ontology-driven and tweet-based hashtags form Hybrid Hashtags. These Hybrid Hashtags are used for tweet topic classification. More details for the approach can be found at [37]. This Hybrid Hashtags approach is used in the framework for processing tweet streams with batch, streaming, and a combination of both approaches.

Next, we show an example of some steps of our proposed approach as in [37]. Suppose we want to classify tweets into two classes of topics: entertainment and sports. We first build an ontology of these class topics. For example, Figure 1 shows a partial ontology of concepts relevant to the class entertainment. As shown in Figure 1, entertainment can be music, which has genre of types pop, opera, and jazz (terminal level). Similarly, entertainment can also be movies, which are productions of the movie industries of Bollywood or Hollywood. The partial ontology of entertainment consists of five types of relationships including hierarchies and 19 nodes, 11 of which are terminal nodes. To be systematic, each of the terminal nodes of the ontology is used as a root concept for generating related hashtags by the Hashtagify tool [38]. We refer to the resulting hashtags as ontology-driven hashtags. For example, the root concept of opera obtained from the ontology of the class entertainment generates 10 related hashtags, each with a corresponding correlation, as a percentage, with the concept opera as shown in the last column of Table 1.

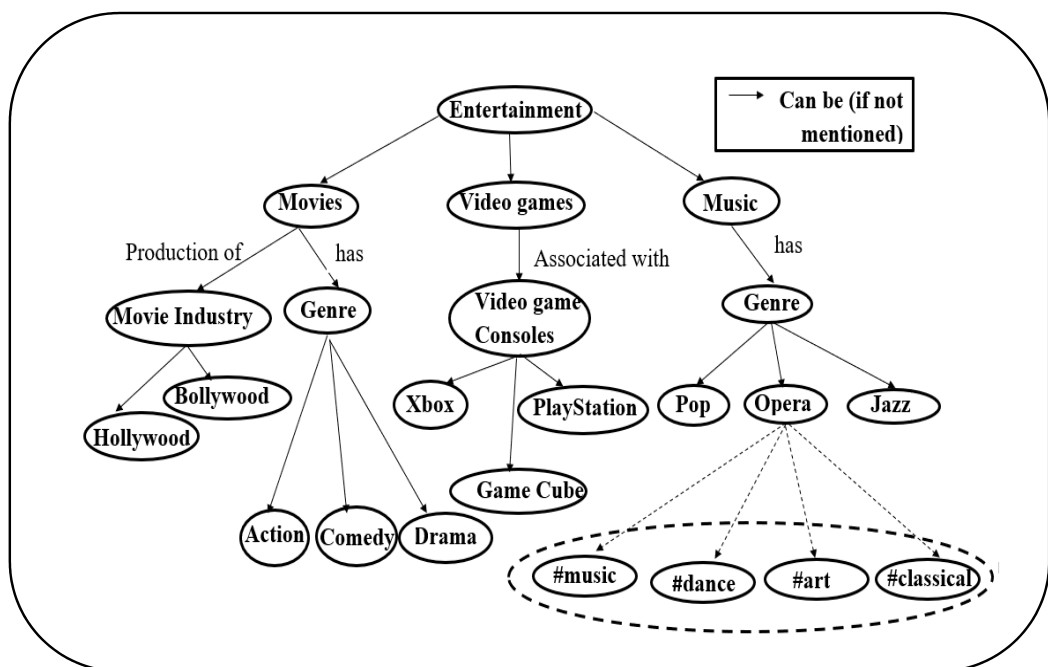

**Figure 1.** Ontology of class concept entertainment.

**Table 1.** Hashtags generated for opera by Hashtagify [32].

| Root Concept | Resulting Hashtags | Correlation with the Root Concept |
|---|---|---|
| | #music✓ | 6.3%✓ |
| | #dance✓ | 2.6%✓ |
| | #art✓ | 2.5%✓ |
| | #classical✓ | 2.1%✓ |
| Opera | #vocal✗ | 1.9% |
| | #Mozart✗ | 1.8% |
| | #Paris✗ | 1.7% |
| | #CGE✗ | 1.7% |
| | #Verdi✗ | 1.6% |
| | #lirica✗ | 1.6% |

If we choose the 4-correlated hashtags (i.e., the top four hashtags with the highest correlation percentages), we obtain #music, #dance, #art, and #classical as the resulting ontology-driven hashtags; the remainder are discarded. These hashtags are shown at the bottom-right-hand corner of Figure 1 with the dashed lines. The same process is repeated for the ontology of the class sports. A total of 78 ontology-driven hashtags are obtained from both classes. These resulting hashtags will be combined with tweet-extracted hashtags using correlation distance as a constraint to establish a set of potentially strong features for tweet classification of the class entertainment. For example, the ontology-driven hashtag #soccer helps us select 10-correlated hashtags extracted from the tweet training dataset, such as #nfl, #football, #premiereleague, #sports, #jersey, and #FIFA.

### 3.2. Apache Storm Architecture

Apache Storm is a real-time distributed stream processing framework [40], shown in Figure 2. The basic components of Storm are spouts and bolts. The data flow between the components is in the form of streams that are an unbounded sequence of tuples (i.e., an ordered list of elements). Spouts are the source of stream data in topology and connected to the raw data source. Bolts are the processing units that process the incoming stream and generate a new stream as output for further processing. A network of spouts and bolts form a topology. A stream processing task is represented as a Directed Acyclic Graph (DAG) where vertices are spouts and bolts, and edges are tuple streams.

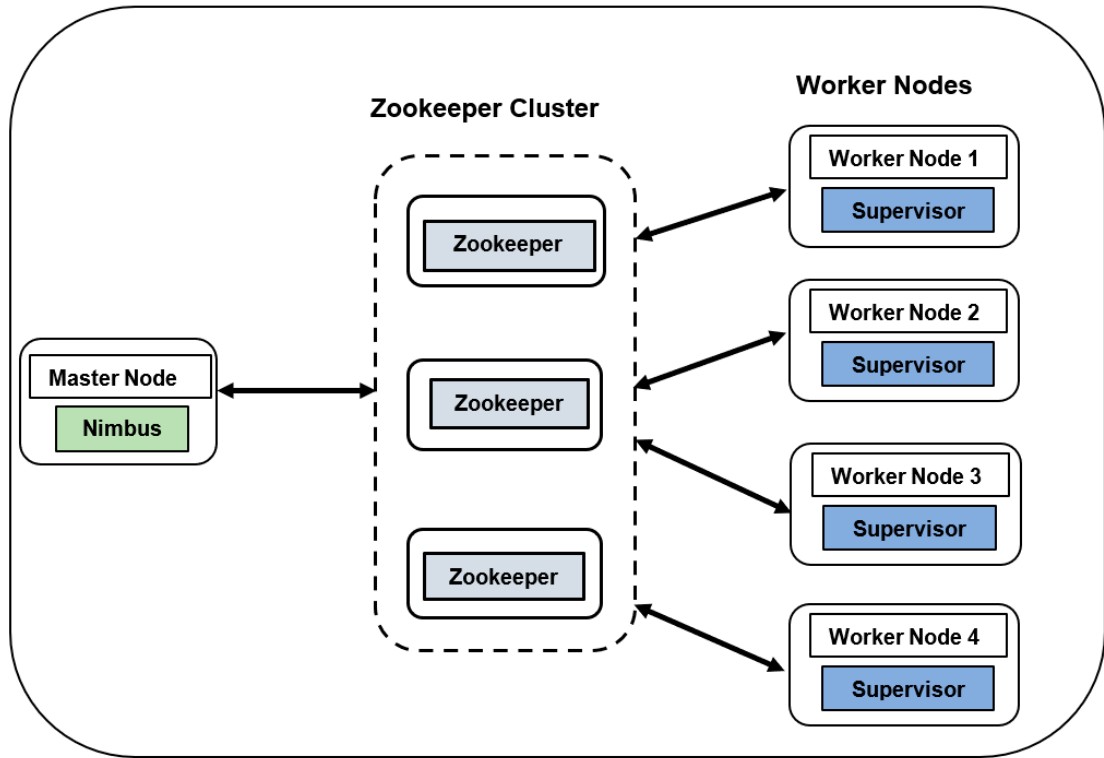

**Figure 2.** Apache Storm architecture.

A user defines the topology according to the application. A high-level Storm architecture (Figure 2) includes three components: Nimbus, Supervisor, and Zookeeper. Nimbus acts as a master node which distributes the computation and coordinates the execution across the worker nodes in the cluster. The topology is submitted to the cluster using Nimbus. Supervisor monitors the worker processes and starts/stop them as necessary. Zookeeper coordinates the communication between Nimbus and Supervisor, monitors their states, and also provides centralized services such as distribution synchronization, group membership, and maintaining configuration information. The actual execution of Storm topology is performed on worker nodes.

Each worker node is associated with one or more worker processes. Each worker process executes a subset of the topology and runs on its own JVM (i.e., Java Virtual Machine) with executors. The executors provide intra-topology parallelism with a different number of instances for each spout or bolt in the topology. Each executor contains multiple tasks that perform the actual data processing. The tasks provide intra-spout/intra-bolt parallelism. A detailed description of Storm can be found in [68].

## 4. Hybrid Hashtag-Based Tweet Topic Classification Framework

We start this section with a formal definition of the task of tweet topic classification using Hybrid Hashtags in our proposed framework. Given a set of tweets $T_b = \{tw_1, tw_2, tw_3, \ldots .., tw_p\}$, where each tweet $tw_i \in T_b$, contains a set of hashtags $H = \{h_1, h_2, h_3, \ldots , h_n\}$ and $|H|$ (i.e., number of hashtags) varies with each tweet. We aim to classify the topics, $y = \{y_1, y_2\}$ where $y_i \in \{sports/entertainment, others\}$ for each tweet $tw_i$, using hybrid hashtags that are derived by the following procedure. We build an ontology graph $OG = \{C, E\}$ first, where vertices $C$ represent the set of concepts and edge set $E$ represents a relationship (*has, associated with, etc.*) between concepts. The terminal node concepts are fed into Hashtagify to produce a hashtag set $H' = \{h_1', h_2', \ldots \ldots ., h_m'\}$, known as concept-based hashtags. These concept-based hashtags are sorted in decreasing order of correlation with the terminal node concepts of the ontology graph (OG) and extract top-k correlated hashtags known as ontology-driven hashtags, $H_{oi} = \{h_{o1}, h_{o2}, \ldots \ldots , h_{ok}\}$ where $H_{oi} \subset H'$. Further, the correlation of $H_{oi}$ with $H$ produces top $k$-correlated hashtags known as tweet-based hashtags, $H_{ti} = \{h_{t1}, h_{t2}, \ldots \ldots .., h_{tk}\}$.

Combination of $H_{oi}$ and $H_{ti}$ forms a set of Hybrid Hashtags $H_{hhi} = \{h_{hh1}, h_{hh2}, \ldots \ldots, h_{hhm}\}$ where each $H_{hhi}$ is associated with each tweet $tw_i \in T_b$. This set of constructed Hybrid Hashtags ($H_{hhi}$) is used as features to build the Hybrid Hashtags-based Tweet topic classifier *(MT)* which is used for further processing in our framework. This part is being implemented on tweet set $T_b$ in batch.

For the online part of the framework, given a stream of tweets $Ts = \{tw_1, tw_2, tw_3, \ldots .., tw_s\}$ where each tweet $tw_i \in T_s$, contains a set of hashtags $H = \{h_1, h_2, h_3, \ldots, h_n\}$, where $|H|$ (i.e., number of hashtags) varies with each tweet. Here, instead of processing a whole batch of tweets $T_b$ at a time, each tweet instance $tw_i \in T_s$ is processed individually for both Hybrid Hashtag construction and classifier building. For each tweet $tw_i$, $MT_b$ first classifies the topic $y_i$ where $y_i \in \{sports/entertainment, others\}$. Then a set of hashtags $H_s' = \{h_1', h_2', h_3', \ldots, h_m'\}$ contained in $tw_i$ are extracted. The correlation of these hashtags ($H_s'$) is first computed with the existing ontology-driven hashtags ($H_{oi}$), to produce the top-k correlated tweet-based hashtags $H_s = \{h_{t1}, h_{t2}, \ldots \ldots .., h_{tk}\}$ for the tweet instance $tw_i$. These tweet-based hashtags ($H_s$) are further combined with $H_{oi}$ to produce Hybrid Hashtags $H_{hhs} = \{h_{hh1}, h_{hh2}, \ldots \ldots, h_{hhm}\}$. This process is repeated for all the tweets in $Ts$. Each time a set of Hybrid Hashtags ($H_{hhs}$) is constructed for the current tweet $tw_{current}$, they are combined with the Hybrid Hashtags generated from all the earlier tweets i.e., $tw_1, tw_2, \ldots \ldots \ldots ., tw_{current}$. Moreover, the existing classifier model $MT_b$ is incrementally updated with new Hybrid Hashtags generated for each new arriving tweet instance.

Figure 3 depicts our proposed Hybrid Hashtag-based Tweet topic classification (HHTC) framework; the upper part shows HHTC-Batch while the lower part shows the HHTC-Stream version of the framework. Both versions are explained in further sections.

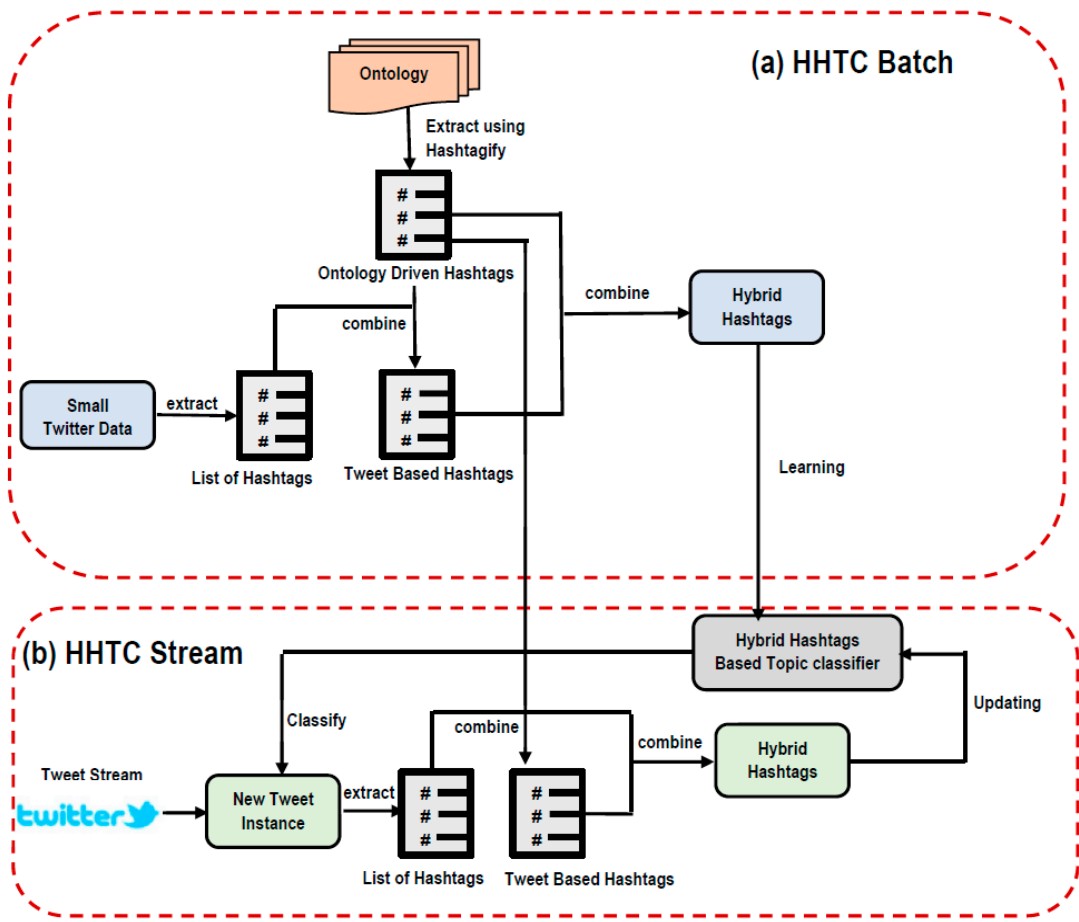

**Figure 3.** Hybrid Hashtag-based Tweet topic classification (HHTC) framework: (**a**) upper part shows the HHTC-Batch version; (**b**) lower part shows the HHTC-Stream version.

### 4.1. HHTC-Batch

The HHTC-Batch version of the proposed framework identifies hashtags using a domain-specific knowledge base (i.e., ontology) in a batch environment, which is similar to our approach in [37]. As shown in Figure 3, we first build a manual ontology of the given class/topic (i.e., entertainment/sports in our case) and extract the ontology-driven hashtags using an automated tool Hashtagify. Then, we start with a small amount of Twitter data collected using the Twitter search API, extract all the hashtags present in the tweets, and combine them with the ontology-driven hashtags to produce $k$-correlated tweet-based hashtags. These tweet-based hashtags are further combined with the ontology-driven hashtags to form Hybrid Hashtags as shown in Figure 3. These Hybrid Hashtags are used as features for building the tweet topic classification model. We provide the code for the implementation in [69].

As HHTC-Batch is used to determine the effectiveness of our Hybrid Hashtags approach, we use the dataset collected through the Twitter search API [37] as the training set and the remainder of the data collected through the streaming API as the test set (More details in Section 5). We build the classification model offline using the training set and then apply it on the test set tweets. Overall, HHTC-Batch helps to identify a good set of hashtag features for tweet topic classification. A major limitation of this approach is that it uses a fixed set of features in the training set to build a classification model, which is not able to adapt to changes in the data stream, and is hence not suitable for real-time analytics. Thus, we propose an online mechanism in the HHTC-Stream version of the framework, which is discussed in the next section.

### 4.2. HHTC-Stream

The HHTC-Stream version of the proposed framework continuously updates Hybrid Hashtags in a stream environment, which is similar to our approach in [39]. As shown in Figure 3, instead of fixed data storage, a continuous tweet stream is generated in this case, which can be processed by online mechanisms. We show the processing of a single tweet instance from the stream in Figure 3, which continuously repeats until there are no tweets remaining for processing. While considering the online mechanism alone, we assume that the Hybrid Hashtags-based Tweet topic classifier model is built on a single instance (i.e., first streaming instance) to simulate the real streaming scenario. As a new tweet instance arrives, it is first classified by the single instance classifier and all the hashtags present in the incoming tweet are extracted. Then, the correlation of this set of hashtags is taken with the existing set of ontology-driven hashtags, to produce k-correlated tweet-based hashtags, which are further combined to produce Hybrid Hashtags for the tweet instance. Using this new set of Hybrid Hashtags, our topic classifier model is updated. Therefore, both the Hybrid Hashtags features and the classification model are updated as new tweet instances arrive over time.

We evaluated the effectiveness of this approach by applying it to the whole dataset (i.e., Twitter data collected from Twitter search and streaming API) instead of having separate training and test sets. A single instance classifier is built initially from the first tweet instance, which is further updated incrementally over time. Each time, a new tweet instance acts as the test set while all other prior instances act as the as training set for the classification model.

The advantage of the HHTC-Stream approach is that it does not use the whole dataset to build the model at once, while it gradually updates the model, and is hence able to adapt to changes in the data, making it suitable for real-time analytics. A major drawback of this approach is that its performance is largely dependent on the arriving data quality due to minimal access to training data initially. As this approach starts with a single instance classifier, and the prediction of the current tweet instance depends upon the classification model built from previous tweets features, the predictions are mostly approximate. To address this issue, we propose a framework combining both batch and online mechanisms in an effective way, which is discussed in the next section.

*4.3. HHTC*

As seen in the above sections, HHTC-Batch and HHTC-Stream have performance advantages in different situations but also have limitations when used alone. The classification model cannot adapt to changes in the former, while the predictions are mostly approximate in the latter. In order to address these problems, we propose a comprehensive HHTC framework in this paper, which not only combines batch and online mechanisms but also provides a generalized and scalable solution.

As our proposed framework is implemented in Apache Storm, both the batch and online mechanisms are implemented together. Here, we use the whole Twitter dataset collected from the Twitter search and streaming API as input to the framework. An initial batch of data processed by the approach explained in Section 3.1 and containing Hybrid Hashtags as features is used to build the initial classifier model. Thus, instead of a single instance classifier, we have a multiple instance classifier trained on the initial batch of data. To exploit the streaming environment, we now apply the online approach to the rest of the data. In particular, every new tweet instance is considered as the test tweet instance, which is first classified by the classifier built on the initial batch of data and then used to update the model with the new Hybrid Hashtags extracted. This process is repeated until the stream ends. Overall, our proposed framework mitigates the individual shortcomings of the batch and online mechanisms. The batch approach helps improve the initial data and model quality, while the online approach improves the model adaptivity over time and provides real-time analytics. In addition, the distributed deployment of our framework in Apache Storm shows that it can be applicable to process large scale Big Data streams, as further discussed in Section 4.3.1.

4.3.1. Distributed Implementation of HHTC in Storm

This section provides a brief overview of the distributed implementation of HHTC using Apache Storm. Both the batch and online approaches for tweet topic classification using Hybrid Hashtags are implemented in Apache Storm. As mentioned in Section 3.2, Apache Storm accepts the stream processing task in the form of a topology whose vertices represent processing units and stream sources, while edges represent the tuple stream. Figure 4 shows the storm topology of our proposed framework, containing one spout and four bolts each with 'N' instances, which shows the amount of parallelism in each spout and bolt. Tweet spouts read the data from the input data files and produce a stream of tweets as output. Since the data stream is represented by the sequence of tuples, tuples emitted by multiple instances of a spout are in the form of *<tweet text, class>*, where *tweet text* contains the raw tweets comprising words, hashtags, URLs, symbols, punctuation, etc., and *class* shows the labels/topic to which the tweet belongs, i.e., entertainment/sports and others in our case.

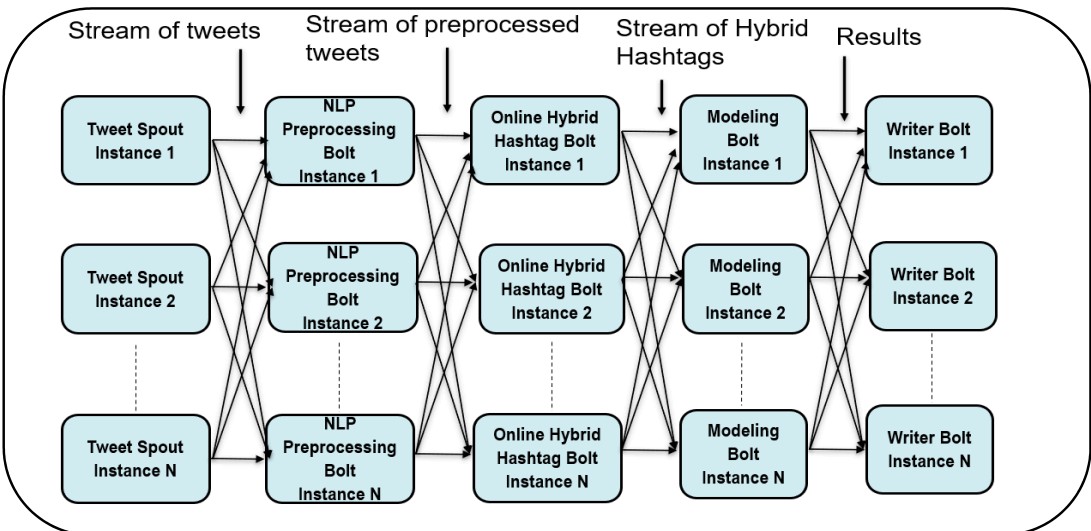

**Figure 4.** Storm Topology of our proposed HHTC framework.

The spout is connected to the NLP (i.e., Natural Language Processing) preprocessing bolt by shuffle grouping as shown in Figure 4. The main function of the preprocessing bolt is to preprocess data using typical natural language processing techniques, such as all characters are converted to lowercase and removal of delimiters, emoticons, white spaces, URLs, and stop words, and stemming. The output tuple stream emitted by the NLP preprocessing bolt is in the form of *<preprocessed tweet text, class>*. The preprocessed text can be extracted words, hashtags, and Hybrid Hashtags. Next is the online Hybrid Hashtag bolt, which takes a stream of preprocessed tweets as input and produces a stream of Hybrid Hashtags as output in the form of *<tweet hybrid hashtags, class>*. The output of the online Hybrid Hashtag bolt is input to the modeling bolt, whose main function is to build the Hybrid Hashtags topic classifier. This bolt not only builds the model but also produces the evaluation results (i.e., accuracy). Finally, the writer bolt writes the results in the output files for further usage.

Now we explain the batch and stream processing part of our framework in Apache Storm. We process the initial batch of data (i.e., the first 8000 tweets) offline using the approach explained in Section 3.1 which is combined with the raw data collected from the Twitter streaming API (i.e., 147,382 tweets). Thus, the initial batch of tweets contains tweet text comprising tweet words, URLs, hybrid hashtags, symbols, punctuation, etc., while the remainder of the data contains the real raw tweets without processing (i.e., tweet words, hashtags, URLs, symbols, punctuation, etc.). For the batch part of the framework, first the batch of data is read by the spouts collectively, which is processed by the NLP preprocessing bolt from which a set of hybrid hashtags are extracted by the online Hybrid Hashtag bolt. Then, a model is built using the batch of data as the training set in the modeling bolt. The stream part of the framework then starts, where tweets are read one-by-one by the spouts; here, two rounds for each tweet instance occurs, one for classifying the new tweet instance and a second to extract the Hybrid Hashtags and update the model. For the first round, the tweet instance read by spouts goes directly to the modeling bolt for classification and the evaluation results are written by the writer bolt. This classified tweet is then again read by the spout and processed by the NLP preprocessing and online Hybrid Hashtag bolts. This time, a new set of Hybrid Hashtags are created and the existing set is updated. Finally, the classifier is updated, which classifies the next tweet instance. This whole process is repeated until there are no tweets in the input stream.

The whole Storm topology is submitted to a distributed cluster for tweet topic classification and the code is provided in [69]. We demonstrate the applicability of our framework on real Twitter data by providing large-scale experimental evaluation results in the further sections.

## 5. Data Collection and Preprocessing

We used data collected through the Twitter search API [37] and Twitter streaming API [70] to evaluate our Hybrid Hashtag approach in the proposed framework. The data collected through the Twitter search API was utilized for the batch part while the Twitter streaming API was used for the online part. The evaluation of hashtag-level topic classification is challenging due to the lack of a "gold standard" labeled dataset. Hence, we used a self-annotation procedure to label the dataset. We collected tweets during four consecutive days from 12–16 August 2019. The data collection process is described as follows. We first created a list of hashtags that are strongly related to the given domain (i.e., sports/entertainment and others in our case). For example, for the sports domain we picked "NFL", "NBA", "football", "Cowboys", "Rio2016". Then we searched in the tweet pool to retrieve tweets containing these hashtags as our seeds. By searching in the tweet pool, we not only retrieved the hashtags being queried but also other hashtags that co-occurred with at least one of the seed hashtags. We considered two classes sports/entertainment and others in our experiments. All the tweets containing hashtags related to sports and entertainment were labeled as sports/entertainment and remainder as others.

We crawled a total of 495,980 tweets following the above process, merged with the old dataset of 8000 tweets collected by the same process in [37] for our experiments. Many tweets were retrieved repeatedly due significant retweeting by users. The detailed raw data collection statistics are shown

in Table 2. After removal of repetitive tweets, a unique, final raw dataset of 289,455 tweets was obtained, of which 271,310 tweets contained hashtags. We analyzed only the tweets containing hashtags, so removed the other tweets from the dataset. Thus, the total number of hashtags in our raw dataset was 1,815,363, containing hashtags related to sports, entertainment, and others domains. As shown in Table 2, the total number of distinct hashtags was 36,049 and the average number of hashtags in each tweet was 6.

**Table 2.** Data collection summary statistics.

| | |
|---|---|
| Number of Tweets | 289,455 |
| Tweets with Hashtags | 271,310 |
| Total Number of Hashtags | 1,815,363 |
| Distinct Hashtags | 36,049 |
| Average Hashtags per tweet | ~6 |

Figure 5 shows the distribution of the number of hashtags in each tweet. As shown, a large number of tweets have a hashtag count in the range of 1–5 but few tweets have hashtags with a count greater than 30. As shown in Figure 5, only one tweet has a hashtag count of 42 and two have hashtag counts of 50. Figure 6 represents the hashtag distribution in our raw tweet collection. It shows the frequency of occurrence of each type of hashtag; for example, the number of hashtags that occur one time, two times, or more. We can observe that the hashtag distribution follows a power law [71], where a small percentage of hashtags has a higher frequency while a large number of hashtags occur in the range of 1–5, as shown in Figure 6. This indicates the diversity of hashtags being generated daily by users and the rapidly evolving nature of tweets. It also shows that some of the popular hashtags are used repeatedly by users. The fact that a large number of hashtags occur infrequently creates a very sparse vector representation of the dataset. This indicates the sparseness of Twitter data, which makes it harder to derive context for tweet topic classification.

Moreover, it is quite challenging and unavoidably labor-intensive to obtains a consistent set of tweets for analytics after crawling due to a large number of retweets, spam tweets, and the noisy nature of tweets. In order to process the data, we first cleaned the dataset by removing URLs, symbols, punctuation, abbreviations, and unnecessary space. We also processed some hashtags by splitting them into smaller segments. Finally, we used stemming to convert the words and hashtags to their original stems.

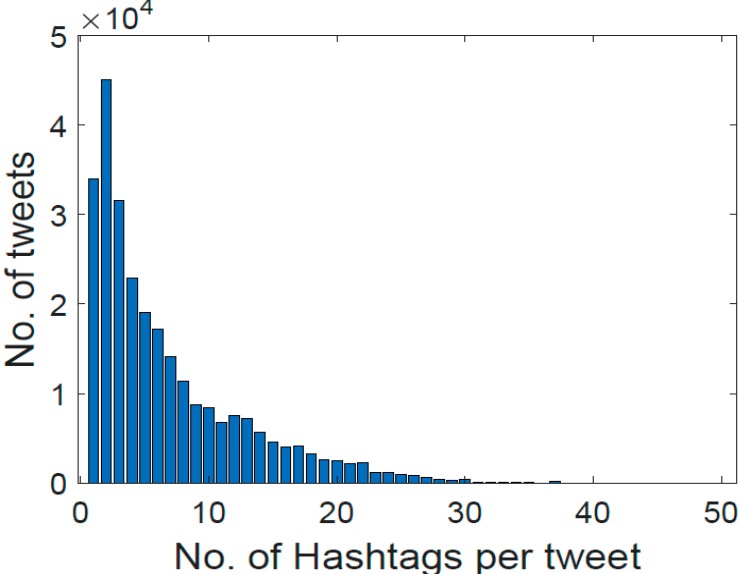

**Figure 5.** Hashtags per tweet.

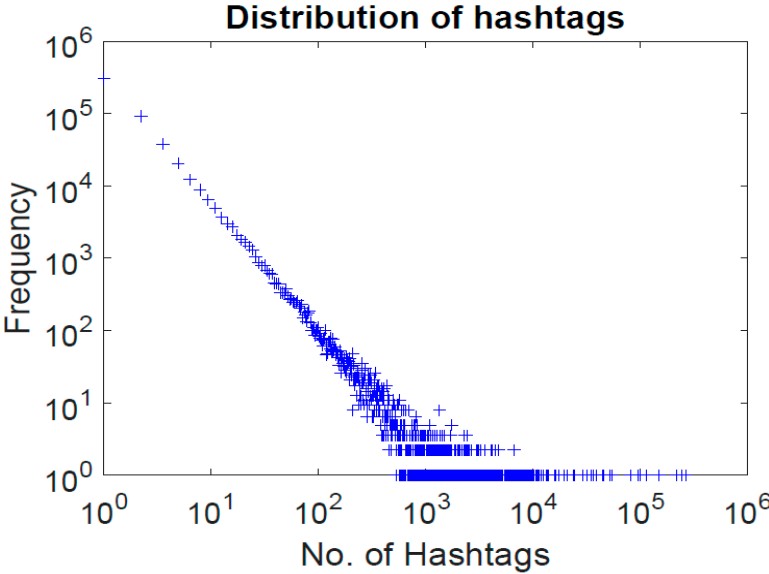

**Figure 6.** Hashtags distribution.

After preprocessing, a total of 155,382 tweets were considered for analysis. We performed experiments in the Storm framework using single and multiple processors to evaluate the effectiveness of our proposed approach in a Big Data stream environment. The Storm cluster was composed of a varying number of virtual machines (VMs or processors) (i.e., 1, 2, 4, and 8) in a system with an Intel Core -i7-8550U CPU 2 GHz processor, 16 GB RAM 8 cores, and 1 TB hard disk. Each of the virtual machines was configured with 4 vCPU and 4 GB RAM. We installed Ubuntu 14.04.05 64 bits OS on each of the VMs with JDK/JRE v 1.8. The Apache Storm version used was 1.1.1 with zookeeper 3.4.9 [68].

## 6. Experimental Results and Evaluation

We evaluated the effectiveness of the Hybrid Hashtags approach for tweet topic classification in the proposed HHTC framework and its two versions HHTC-Batch and HHTC-Stream. In HHTC-Batch, the old dataset of 8000 tweets collected using the Twitter search API [37] was used as a training set and the remaining large dataset of 147,382 tweets collected using the Twitter streaming API [70] was used as a test set. In HHTC-Stream, the whole dataset of 155,382 tweets was used without any separate training and test set. The first instance was used as the training set initially to start processing, which was incrementally updated with each new tweet instance, while each new tweet instance was considered to be in the test set. In HHTC, the old dataset of 8000 tweets collected using the Twitter search API [37] was used as a training set to build an initial model, which was further updated with each new tweet instance, while each new tweet was considered as a test set.

We compared the results of all versions of the framework using four experimental cases: (1) tweet words only (i.e., using only tweet words as features), (2) tweet words and hashtags (i.e., using both tweet words and hashtags as features), (3) tweet hashtags only (i.e., using only hashtags contained in tweets as features), and finally (4) our hybrid hashtags approach (i.e., using hybrid hashtags as features).

### 6.1. Classification Performance

The tweet topic classification performance of the proposed *HHTC* framework and its two versions are compared in this section. We evaluate the classification performance using an accuracy measure which is defined as the ratio between the number of test set tweets classified correctly and the total number of test set tweets. The average accuracy for test set tweets with 10 times cross-validation are reported for HHTC-Batch while average accuracy over all instances is reported for HHTC-Stream and HHTC. We applied naïve Bayes [72], SVM [73], and *k*-NN (i.e., *k*-nearest neighbor) algorithms for classifying tweet topics for all four experimental cases in HHTC-Batch, while naïve Bayes, Hoeffding

trees, and Hoeffding adaptive trees were used for HHTC-Stream and HHTC. Table 3 shows the average classification accuracy obtained for HHTC-Batch.

**Table 3.** Comparison of HHTC-Batch Tweet topic classification results with different sets of features.

| Features | Naïve Bayes | SVM | *k*-NN |
|---|---|---|---|
| Words Only | 74.7% | 71.0% | 58.2% |
| Words and Tweet Hashtags | 91.1% | 87.4% | 73.3% |
| Tweet Hashtags Only | 85.2% | 92.3% | 87.0% |
| Hybrid Hashtags | 96.4% | 95.3% | 92.0% |

Looking at the outcome of Table 3, we observe that the average accuracy using Hybrid Hashtag features outperforms compared with other types of feature sets for all three classification algorithms. This shows that our approach to determine the strong set of hashtags as features is robust to classification techniques. It also shows that the quality of the training set affects the classification performance substantially since we use a fixed set of 8000 tweets with Hybrid Hashtags as features to classify a large number of tweets in the test set. As shown in Table 3, naïve Bayes performance is best for all approaches except for tweet hashtags since it is widely used in text classification and robust to irrelevant features. Using words alone performs worst for all the algorithms since they increase the noise in tweets and make it harder to derive relevant topics. Most of the words increase relevance for the topics by combining with the hashtags, which is why the words and tweet hashtags approach performance is better than using the words only approach. Using tweet hashtags as features performs closest to the Hybrid Hashtag approach for SVM and k-NN. Thus, hashtags improve classification and ontology-based hashtags further enhance it. These results also show that our Hybrid Hashtag approach can be used to build a model with a limited amount of labeled training data.

To evaluate the effectiveness of our proposed Hybrid Hashtags approach, we compared it with state-of-the-art tweet topic classification techniques in the HHTC-Batch version of the framework as shown in Table 4. To compare the performance of our approach, we extracted hashtags contained in each tweet in the training set and applied each technique shown in Table 4, which produced another set of hashtags from the original set from the raw data. This new set of hashtags were used as features to build a classification model using the naïve Bayes algorithm [72], which was further applied to the test set with 10-times cross validation to produce average accuracy.

**Table 4.** Comparison of our proposed approach with state-of-the-art tweet topic classification techniques.

| Technique | Accuracy |
|---|---|
| LDA Topic Model | 87.0% |
| LIWC Model | 62.4% |
| Wordnet | 85.0% |
| Wikipedia | 86.8% |
| Wordnet + Wikipedia | 89.0% |
| LDA Topic Model + Ontology Driven Hashtags | 91.0% |
| Hybrid Hashtags | 96.4% |

Our proposed Hybrid Hashtags approach is compared with the lexicon-based approach LIWC (Linguistic Inquiry and Word Count) [74,75], approaches using external lexical resources such as WordNet [48], Wikipedia [20], hybrids of multiple resources [52] and LDA topic modelling [76]. As shown in Table 4, our proposed technique performed best, while the LIWC lexicon-based model showed the worst classification performance among other techniques. LIWC is a text analysis algorithm that uses the percentage of words relevant to the predefined categories to compute semantics of text; however, due to the frequent evolution and casual expression of tweet hashtags, most hashtags are not categorized by the LIWC dictionary, leading to a poor classification model.

The latent Dirichlet allocation (LDA) is an unsupervised technique to cluster words in semantically similar groups known as topics. These models have been successfully used in the clustering of a variety of long and short documents [76]. We applied LDA to our tweet hashtags dataset with 10 topics and produced quite relevant clusters of hashtag topics, yielding 87% accuracy. For further improvement, we combined LDA topic hashtag features with the ontology-driven hashtags generated by Hashtagify and built a classification model that improved accuracy by 4.6%, as shown in Table 4. Nearly equal classification performances were produced when WordNet and Wikipedia were used as individual lexical resources for tweet topic classification as in [20,47], and accuracy was improved by combining them, similar to the approach taken in [52], due to the broader coverage of the lexical database. Overall, the results show that our Hybrid Hashtag approach is effective and can be used to improve tweet topic classification.

Table 5 shows the classification results obtained for HHTC-Stream. As shown, using Hybrid Hashtags as features results in better performance than that of all other types of feature sets. The online preprocessing and analytics perform slightly better than the batch analysis for all approaches using naïve Bayes. This might be due to the gradual adaptation of the classifier with more tweets as compared to the fixed set of features in the batch analysis. Compared to HHTC-Batch, the classification performance using only words as features and using both words and hashtags as features is improved by 6.2% and 3.5%, respectively, in HHTC-Stream. This could be due to incremental adaptation of the classifier with new words and hashtags in HHTC-Stream compared to the fixed model with a fixed set of features in HHTC-Batch. There is a slight increase of 0.8% in accuracy for our hybrid hashtags approach since most representative Hybrid Hashtag features related to class/topic are identified initially in the batch analysis, which will also not change significantly in the incremental streaming analysis. Using words only still performs worst in all cases, while words with hashtags improve the performance. The performance in the case of using only tweet hashtags as features perform closer to the Hybrid Hashtags approach for Hoeffding adaptive trees and Hoeffding trees, which are incremental anytime decision trees capable of learning from data streams.

**Table 5.** Comparison of HHTC-Stream Tweet topic classification results with different sets of features.

| Features | Naïve Bayes | Hoeffding Trees | Hoeffding Adaptive Trees |
|---|---|---|---|
| Words Only | 79.3% | 55.5% | 57.8% |
| Words and Tweet Hashtags | 94.3% | 71.5% | 76.4% |
| Tweet Hashtags Only | 89.8% | 75.3% | 91.0% |
| Hybrid Hashtags | 97.2% | 80.0% | 92.2% |

The classification performance of our proposed HHTC framework is shown in Table 6. Our hybrid hashtag approach is still the best performing. Moreover, the accuracy of all three algorithms is better than HHTC-Stream with all approaches. This might be due to the fact that the initial classifier model provides ample feature learning which improves gradually with the streaming analysis. For our Hybrid Hashtag approach, initial batch analysis selects the most representative features that will not change significantly in further analysis, hence resulting in slight improvements of 0.31%, 0.25%, and 2.49%, as shown in Tables 5 and 6. Compared to HHTC-Stream, the adaptation of the classifier is faster in HHTC since the initial classifier model is built with more training set features. Overall, the results show that the combination of batch and streaming approaches performs better than batch and streaming individually for tweet topic classification, which shows the applicability of HHTC to real tweet streams.

**Table 6.** Comparison of HHTC Tweet topic classification results with different set of features.

| Features | Naïve Bayes | Hoeffding Trees | Hoeffding Adaptive Trees |
|---|---|---|---|
| Words Only | 81.0% | 56.0% | 60.7% |
| Words and Tweet Hashtags | 94.6% | 76.1% | 78.3% |
| Tweet Hashtags Only | 90.0% | 77.8% | 88.5% |
| Hybrid Hashtags | 97.5% | 80.2% | 94.5% |

*6.2. Execution Time*

In this section, we compare the total execution time for preprocessing and analytics for the proposed framework HHTC and its two versions. We consider the single processor execution time for the naïve Bayes algorithm in this experiment.

Table 7 compares the execution time for all approaches. It shows that our Hybrid Hashtag approach has the shortest execution time compared to other approaches. This is due to the fact that the number of features is reduced drastically when we use only hashtags as features, which reduces the time needed to preprocess and apply machine learning. The tweet hashtag approach execution time is closest to ours, while the words and tweet hashtag approach takes the most time to execute due to the large feature set. As shown in Table 7, HHTC-Stream has the shortest execution time for all approaches, while HHTC has the longest execution time. The rationale behind this is that HHTC initially takes a batch of data to develop a model which increases the overall execution time, while HHTC-Stream continuously processes and updates the model, which reduces the execution time. Finally, although HHTC classification performance is better than that of other versions, the overall execution time is much higher. Thus, there is a tradeoff between the accuracy and execution time, and usage of HHTC depends upon the application requirements. Furthermore, HHTC-Stream has the advantage of having the shortest execution time and acceptable classification accuracy, which also makes it suitable for real-time analytics.

**Table 7.** Total execution time (in seconds) for both preprocessing and classification combined.

| Features | HHTC-Batch | HHTC-Stream | HHTC |
|---|---|---|---|
| Words Only | 12.14 | 3.20 | 40.24 |
| Words and Tweet Hashtags | 13.37 | 3.40 | 45.19 |
| Tweet Hashtags Only | 12.07 | 2.71 | 36.15 |
| Hybrid Hashtags | 11.50 | 2.56 | 35.08 |

*6.3. Scalability and Speed*

We investigated the scalability and speed of our approach by conducting experiments in a distributed environment using Apache Storm. The scalability was evaluated using throughput and execution time. Throughput is the total number of data points (i.e., tweet instances) processed per unit time (i.e., seconds).

Here we report the results using Hybrid Hashtags as features for both HHTC-Stream and HHTC to evaluate its scalability. As shown in Figure 7, the throughput for HHTC-Stream is higher than HHTC in every case with the number of processors (i.e., 1, 2, 4, 8), which is due to the faster and continuous processing of HHTC-Stream compared to the slower processing of HHTC in the batch phase. Since the rate of processing also depends upon the number of features in the training set, the greater the number of features, the longer the processing time; thus, the processing rate is slower, leading to lower throughput. That is why throughput is higher for the Hybrid Hashtag approach compared to other approaches, that is, due to fewer features and a faster rate of processing.

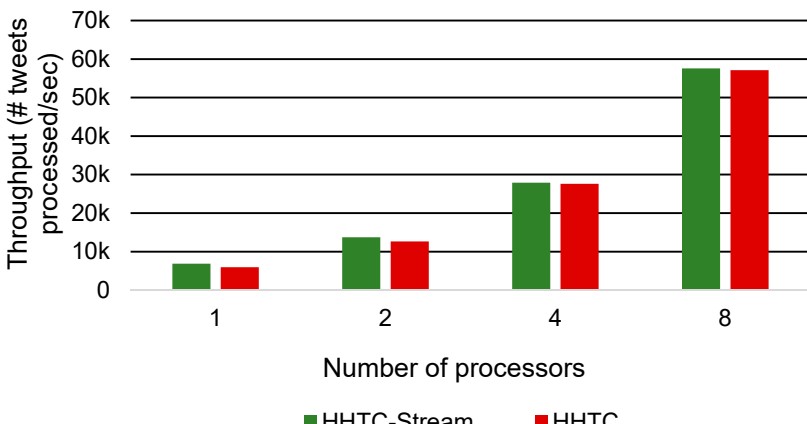

**Figure 7.** Comparison of throughput with varying number of processors for both HHTC-Stream and HHTC where Throughput is the number of tweets (in thousands) processed per second.

Table 8 presents the variation in execution time with different dataset sizes. This experiment was performed to evaluate the scalability of our Hybrid Hashtags approach. To perform this experiment, we divided the original dataset into smaller chunks shown as a fraction of the dataset. The fraction of the dataset (i.e., 0.2, 0.4, 0.6, 0.8) is the proportion of data considered out of the original dataset (i.e., 1). As shown in Table 8, execution time increases almost linearly as the dataset size increases, which shows that our proposed approach is scalable and can be applied to large data streams.

**Table 8.** Execution time (in seconds) with varying dataset sizes for both HHTC and HHTC-Stream to evaluate the scalability of our Hybrid Hashtags approach.

| Fraction of Dataset | 0.2 | 0.4 | 0.6 | 0.8 | 1 |
|---|---|---|---|---|---|
| Dataset Size | 31,076 | 62,153 | 93,230 | 124,306 | 155,382 |
| HHTC | 25.15 | 25.83 | 30.34 | 31.06 | 35.08 |
| HHTC-Stream | 1.06 | 1.19 | 1.50 | 2.11 | 2.56 |

Finally, we estimated the effect of the number of computing nodes on the execution time for Storm implementation. This experiment was performed to evaluate the speed of our approach with an increasing number of processors in a distributed environment. We tested four different cluster configurations with 1, 2, 4, and 8 nodes.

Tables 9 and 10 present the speed results for HHTC-Stream and HHTC, respectively. We observe that the execution time decreases as we add more processors to the cluster, with the shortest execution time achieved with eight processors. This trend can be seen for all of the approaches. The rationale behind the decrease in execution time with increase in parallelism is that the same dataset is being processed in parallel by distributing it to multiple processors, which reduces the execution time substantially.

**Table 9.** Speed up for HHTC-Stream (in seconds) with different numbers of nodes for different sets of features.

| Execution Time | | | | |
|---|---|---|---|---|
| Number of Processors | 1 | 2 | 4 | 8 |
| Words Only | 3.20 | 2.02 | 2.23 | 1.51 |
| Words and Tweet Hashtags | 3.40 | 2.86 | 2.64 | 1.62 |
| Tweet Hashtags Only | 2.71 | 1.92 | 1.62 | 1.61 |
| Hybrid Hashtags | 2.56 | 1.90 | 1.67 | 1.34 |

**Table 10.** Speed up for HHTC (in seconds) with different numbers of nodes for different sets of features.

| | Execution Time | | | |
| --- | --- | --- | --- | --- |
| **Number of Processors** | **1** | **2** | **4** | **8** |
| Words Only | 40.24 | 33.13 | 24.59 | 16.22 |
| Words and Tweet Hashtags | 48.19 | 45.32 | 30.76 | 28.12 |
| Tweet Hashtags Only | 36.15 | 18.92 | 14.28 | 11.06 |
| Hybrid Hashtags | 35.08 | 18.38 | 11.12 | 10.76 |

The scalability and speed results show that our approach is efficient, scalable, and robust, and therefore appropriate for Twitter Big Data stream topic classification.

**7. Conclusions and Future Work**

Tweet topic classification using hashtags is a challenging task due to frequent evolution and non-standardized expressions (i.e., ambiguous, slang, abbreviations) of hashtags by users. Existing approaches, using external lexical resources for deriving semantics from hashtags, suffer from the issues of limited coverage and lack of scalability and generality. In this paper, a novel Hybrid Hashtags approach is presented. The approach derives semantically relevant hashtags using a domain-specific knowledge base of topic concepts. We first evaluate the effectiveness of the proposed HHTC-Batch environment where we compare results of our approach with other state-of-the-art tweet topic classification techniques. The results show that our proposed technique is effective for tweet topic classification and can be used in combination with some existing techniques to build a classification model, even from a limited labeled training data set. Specifically, we propose a technique that combines tweet hashtags with enhanced ontology-driven hashtags and show by experiments that the technique is effective. Further, to deal with large data streams, an online technique (HHTC-Stream) is presented and implemented in a real-time distributed framework, Apache Storm. Finally, a comprehensive framework, Hybrid Hashtag-based Tweet topic classification (HHTC) is presented where we effectively combine both batch and online mechanisms. Experimental results show that the combination of batch and online mechanisms helps to improve classification performance but takes longer to execute. However, the online approach drastically reduces the execution time, which is helpful for real-time analytics applications. Thus, there is a tradeoff between speed and accuracy. Overall, HHTC-Stream is beneficial for faster application requirements while HHTC-Batch and HHTC can be used for more accurate model-building requirements. Both HHTC-Stream and HHTC improve throughput, speed, and execution time in a multiprocessor environment. Although we studied a binary tweet topic classification problem in this work, the proposed approach is also general enough to be applicable to non-binary classification. In theory, like many machine learning techniques, having a large number of classes should not impact the accuracy, if the training data consists of a well-balanced class distribution that is large enough to derive valid results. However, in practice, this may not be easy to obtain due to the rarity of the use of some hashtags. Thus, to deal with such a situation in practice requires a large set of labeled data. In our future work, we will apply our proposed technique to multi-class classification problems and further evaluate it with more diverse datasets.

**Author Contributions:** Methodology, V.G. and R.H.; software, V.G.; formal analysis, R.H.; writing—original draft preparation, V.G.; supervision, R.H.; writing—review and editing, R.H. and V.G. All authors have read and agreed to the published version of the manuscript.

**Funding:** This research received no external funding.

**Conflicts of Interest:** The authors declare no conflict of interest. Any opinions, findings and conclusions or recommendations expressed in this material are those of the author(s).

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
