# Peer review of "Real-Time Tweet Analytics Using Hybrid Hashtags on Twitter Big Data Streams†"

_information, doi:10.3390/info11070341_

Round 1

Reviewer 1 Report

This paper presents a solid work for tweet topic classification. The existing literature is well studied. Material and methods are described in detailed (some improvements are needed), and extensive experiments and results are presented. Please find my comments below:

Writing issues

  • There are some long sentences that are hard to follow; example: “Although these approaches are applicable for classifying tweet sentiments that can be easily described by the semantics of keywords or hashtags but not for general topic classes which require a diverse set of hashtags to cover various aspects of it.”
  • There are sentences that are unclear; example: “Although ontologies are widely used for natural language generation, extracting semantics from unstructured texts and intelligent information integration [29], we apply it to extract domain-specific or Ontology-Driven Hashtags to cover various aspects of a general topic/class.”
  • Some informal writings forms are used that should be changed to formal form (don’t —> do not, that’s —> that is, etc.)
  • Inappropriate usage of “it’s” in the discussion section (you may want to  change them to “its” as they show possession)
  • Add comma after e.g. and i.e.
  • I strongly recommend an end-to-end proofreading before the next submission. 
  • Improve quality of images and diagrams (some of them are of low resolution)
  • Captions: captions for figures and tables need to be improved. More detailed descriptions are needed to make them as self explanatory as possible (i.e., without referring to the text, a reader should be able to grasp the main idea or purpose of a figure/diagram.)

Algorithm and Method:

  • Section 5 (Data): how did you label each tweet? By human subject? Or only by checking if it contains a certain hashtag? This has not been clarified in the paper. 
  • This reviewer found the explanation of the proposed framework and its two variations a little unclear. More concrete explanations would help a potential reader to follow your design. Also, in Figure 3, you may want to better separate different components of design that belong to each variation (e.g., by using different color codes, bounding box, etc.).
  • The setup of classification task is not clear. For instance, authors can add a single paragraph explaining what is input, what is process, and what is output. As input, you can say a tweet will be represented as a vector of terms/hashtags. The process will be your proposal, and the output could be a class label. A formal description would be very useful.
  • Is your classification based on three classes (including “other” as a separate class label) or two classes? This wasn’t clear in the paper.
  • How would you expect that increase in number of classes will impact accuracy? Any insights would be useful for a potential reader; as in practice, we would expect to deal with a larger number of classes. 
  • Execution time: It would be useful to know how long would it take for a single tweet to be classified in test mode, based on different solutions.
  • Do you have any public repository for your codes (say on GitHub) for the purpose of reproducibility? If so, please provide a link in the paper. If not, please do it (as it would give other researchers this opportunity to compare their work against yours).

Reviewer 2 Report

After the first revision, the manuscript has been improved. However, there are still some points in the paper are not clear, which are addressed in the follows:

  1. The advantages and the importance of this work should be further discussed.

  1. Beside the Fig.1, it would be good to have an overview of the proposed framework (preferably a system diagram or flowchart), to give readers a quick understanding of the framework.

  1. Author should analyze how to set the parameters in the framework (section 6). Do they have the “optimal” choice?

  1. Some new references related to data science may be useful for improving this manuscript, such as [1] IEEE Transactions on Industrial Informatics, 16, 8, 5327-5334, 2020. [2] Physical Review E 91 (1), 012801, 2015. [3] New Journal of Physics, 21, 015005, 2019. [4] IEEE Transactions on Emerging Topics in Computational Intelligence, 2(3), 214-223, 2018. [5] Physica A , 542, 123514, 2020.

  1. The quality of some figures is poor, such as fig 5 and 6. Please enlarge the caption and enhance the resolution.

  1. The language should be improved further.
